# Impact of Artificial Nutrition on Postoperative Complications

**DOI:** 10.3390/healthcare8040559

**Published:** 2020-12-14

**Authors:** Sergio Sandrucci, Paolo Cotogni, Beatrice De Zolt Ponte

**Affiliations:** 1Surgical Oncology Unit, City of Health and Science, University of Turin, 10126 Turin, Italy; beatrice.dezoltponte@edu.unito.it; 2Pain Management and Palliative Care, Department of Anesthesia, Intensive Care and Emergency, Molinette Hospital, University of Turin, 10126 Turin, Italy; paolo.cotogni@unito.it

**Keywords:** nutritional screening, enteral nutrition, parenteral nutrition, immunonutrition

## Abstract

Malnutrition is common in surgical cancer patients and it is widely accepted that it can adversely affect their postoperative outcome. Assessing the nutritional status of every patient, in particular care of elderly and cancer patients, is a crucial feature of the therapeutic pathway in order to optimize every strategy. Evidence exists that the advantages of perioperative nutrition are more significant in malnourished patients submitted to major surgery. For patients recognized as malnourished, preoperative nutrition therapies are indicated; the choice between parenteral and enteral nutrition is still controversial in perioperative malnourished surgical cancer patients, although enteral nutrition seems to have the best risk–benefit ratio. Early oral nutrition after surgery is advisable, when feasible, and should be administered in all the patients undergoing elective major surgery, if compliant. In patients with high risk for postoperative infections, perioperative immunonutrition has been proved in some ways to be effective, even if operations including those for cancer have to be delayed.

## 1. Introduction

Surgery is currently the main therapeutic approach to many solid neoplasms; surgical techniques have considerably evolved in the last 20 years, leading to an increase in overall survival and quality of life [1]. However, surgery cannot provide significant effects if systemic conditions are not taken into account. Nutritional status, for instance, has been demonstrated to have a major impact on postoperative complications and the length of hospital stay [2], even in a minimally invasive setting. Chemotherapy, radiotherapy, cancer progression and host response to the disease can contribute to adverse effects of malnutrition [3,4,5]. Screening tools and hystotipe can affect the incidence of malnutrition [6]; in any case, malnutrition has been associated with poor prognosis and quality of life in any cancer type. The modifications of dietary intake and metabolism induced by cancer and surgery can be counteracted by nutrition therapy, whose efficacy may be optimized through synergy with physical activity [2]. Weight loss, stress and systemic inflammation in surgical oncology patients are all independently associated with a poor prognosis, increased postoperative morbidity resulting in interruptions of postoperative anticancer treatments, and reduced quality of life. Given the high incidence of hypo or malnutrition or metabolic alterations among cancer patients, to implement early interventions against all relevant impairments appears reasonable. (Table 1).

Perioperative malnutrition remains largely underdiagnosed and untreated, despite 24–65% of patients undergoing surgery are at risk of malnutrition or undernutrition [7]. A minority of hospitalized patients are actually screened for malnutrition; in the US, only 20% of hospitals adopt a nutrition screening protocol for surgical oncology patients [8].

Data from the National Surgical Quality Improvement Program (NSQIP) demonstrate that malnutrition is among the first 10 preoperative risk factors leading to a poor outcome or increased mortality [9]; moreover, among the main causes of postoperative mortality malnutrition is the only identifiable and modifiable factor. It is estimated that around 30% of patients are malnourished at hospital admission, and the Healthcare Cost, and Utilization Project (HCUP) indicates that only 3% of these patients are properly identified during their hospitalization [10]. The HCUP project reveals also that fewer than 7% of malnutrition-related hospital stays are submitted to specific and significant nutritional interventions.

## 2. Material and Methods

A PubMed/Medline search was performed using search terms “nutrition and surgery,” “nutrition and surgical oncology”, “effects of nutrition on surgical complications,” “nutritional screening in surgical oncology patients”. Duplicates were removed and articles were screened for relevance. The extreme heterogeneity of studies did not allow a systematic review of retrieved data and a descriptive review was thus made, discussing some points and guidelines of relevant professional associations.

## 3. Results

### 3.1. The Metabolic Response to Immobilization and Surgical Trauma

To assess and optimize the nutritional status of patients undergoing surgery, it is essential to understand how the metabolism changes during injury, and why a poor nutritional status is a risk factor for postoperative complications (Figure 1).

Surgical trauma induces hormonal, hematological, metabolic, and immunologic changes aimed to counteract the state of stress that challenges metabolic and physiologic equilibrium [10,11]. The stress response and tissue injury are strictly related; an enhanced stress reaction can lead to major adverse consequences including catabolism, hyperglycemia and immunosuppression (Figure 2).

The goal of a nutritional intervention is to attenuate catabolism while preserving the processes of the surgical stress. To reach a complete restitutio ad integrum, this metabolic response is necessary, but when the inflammatory and stress response are prolonged, this may require nutritional therapy. The negative effect of long-term protein and calories deficiency on the outcome of critically ill surgical patients has been recently demonstrated [11].

Systemic Inflammatory Response Syndrome (SIRS), that is the cytokine response to infection and injuries, has a strong impact on metabolism, since it leads to catabolism of glycogen, protein, and fat with release of glucose, free fatty acids, and amino acids shifted from their normal function of maintaining the peripheral protein quote (especially muscle) to the tasks of wound healing and immune response [12,13].

The loss of lean mass is the direct consequence of protein catabolism. Nutritional therapy in the immediate postoperative phase may only minimally counteract muscle catabolism. In catabolic surgical patients, interventions to preserve functional capacity and correct malnutrition must be applied preoperatively to preserve functional capacity [14]. Multidisciplinary strategies can promote earlier patient recovery from major surgery mitigating the metabolic response to surgical stress [15].

### 3.2. The Patient at Risk and Nutritional Assessment

Patients undergoing oncologic gastrointestinal surgery are at risk of malnutrition as a result of anorexia, intentional fasting, malabsorption or diarrhea [6,16,17]. For these reasons nutritional screening is advisable, even when the nutritional risk is not overtly present, at diagnosis, after a month and again, 6 months later. However, a global consensus concerning the optimal method for nutritional risk screening and the assessment of nutritional status does not exist.

To stress the close interaction between nutritional status and disease, the term “disease-related malnutrition” (DRM) has been proposed [18]. The World Health Organization has defined as “Disease Related Malnutrition” (DRM) a body mass index (BMI) <18.5 kg/m^2^ [19], although disease-related weight loss in overweight patients is not necessarily associated with low BMI values. It is important to keep in mind that patients undergoing surgery, especially cancer patients, may have a “metabolic risk” related to chronic low-grade inflammation, weight loss and changes in body composition, even if they are not underweight [20].

The recently released ESPEN (European Society for Clinical Nutrition and Metabolism) Endorsed Recommendation, proposed the GLIM (Global Leadership Initiative on Malnutrition) criteria which combine phenotypic items (weight loss, reduced BMI, and reduced muscle mass) and etiologic items (reduced food intake/assimilation and disease burden/inflammation). GLIM recommends that the combination of at least one phenotypic criterion and one etiologic criterion is required for malnutrition diagnosis [20] 

Albumin, prealbumin, and transferrin have for a long time been considered as determinants of perioperative nutritional status [2]; in particular, in colorectal surgery preoperative hypoalbuminemia has been related to postoperative complications, such as septic shock, surgical site infections, and pneumonia [21,22]. A systematic review on general surgery patients (>65 years) has shown that weight loss and serum albumin concentration can be predictive for postoperative adverse outcomes [23]. 

However, serum albumin levels have a poor correlation, with nutritional status being influenced by a variety of factors. Albumin synthesis can decrease during surgery and increase with the onset of inflammatory response; the concentration of plasma albumin observed after surgery is a consequence of its synthetic rate and not of nutritional status, and can be influenced by its relatively long half-life (approximately 14 to 20 days or its redistribution as a result of increased capillary permeability secondary to inflammation or infection [24].

Despite these considerations, nutritional scoring systems based on serum albumin levels and other predictors such as lymphocyte count, serum cholesterol, neutrophil: lymphocyte ratio, or lymphocyte: monocyte ratio are widely used.

The CONtrolling NUTritional status (CONUT) score consists of three parameters, serum albumin, cholesterol, and total leucocyte count (TLC); it has been used to assess the nutritional status in upper GI (gastrointestinal) surgery [25,26].

The PNI (Prognostic Nutritional Index) is calculated using the serum albumin concentration and the total leucocyte count and has been tested as a prognostic predictor in digestive oncologic surgery [27,28].

The NRI (Nutritional Risk Index) relies on serum albumin concentration and the ratio of actual to usual weight: the NPS (Naples prognostic score) is a composite score based on albumin and cholesterol concentrations, neutrophil: lymphocyte ratio, and lymphocyte:monocyte ratio [29].

Although a recent study showed that CONUT and PNI can help in assessing the risk of severe postoperative complications [30], specificity and sensitivity of CONUT, PNI, NPS, and NRI as undernutrition assessment tools can be biased by the use of albumin levels’ assessment, a well-known risk predictor in a broad sense rather than a marker of undernutrition [24]. The albumin-based tests can thus be considered more valuable as prognostic indexes.

To evaluate the nutritional status of patients with cancer. The Patient Generated Subjective Global Assessment [PG-SGA] has been proposed [31]. The scored PG-SGA yields a numerical score joint with a global rating of well-nourished (SGA-A), moderately nourished/suspected malnutrition (SGA-B), or severely malnourished (SGA-C). The score can be used to triage nutrition intervention, while the classification can determine the nutritional status, and dynamically measure a change in nutritional status, or in the quality of life [32]. PG SGA has shown its efficacy to assess the nutritional status in patients undergoing colorectal surgery and also liver surgery for HCC (hepatocarcinoma) [33,34]. In a recent study PG-SGA has been compared with NRI, showing a better efficacy in terms of sensitivity and specificity in predicting postoperative complications (98% vs. 72%) [35].

The metabolic risk associated with disease-related malnutrition (DRM) can also be detected by the Nutritional Risk Screening-2002 (NRS-2002) which assesses BMI, unintentional weight loss, changes in food intake, and the disease severity (low, moderate, or severe), adjusted for age >70 years [36]. NRS-2002 and PG-SGA are considered similar for assessing nutritional status, although the latter seems to be more suitable for detecting malnutrition in patients with cancer [3].

The Malnutrition Universal Screening Tool (MUST) can score the acute effect on nutrition considering BMI and unintentional weight loss (UWL). This latter is considered an important predictor of malnutrition risks even if BMI is not pathologic, and has been associated with poor surgical outcomes in patients with advanced cancer [37]. The value of UWL as a poor prognosis predicting factor is not clarified as existing studies are biased by confounding factors such as emergency surgery and performance status [38].

MUST is a simpler tool compared with SGA and, compared with NRI, has a good predictive value and a higher sensitivity, specificity, positive and negative predictive value of the relationship between malnutrition and the length of hospital stay, mortality, and morbidity [37].

### 3.3. Nutritional Interventions in Surgical Patients

Nutritional therapy in patients undergoing surgery is aimed to correct undernutrition before surgery and maintain nutritional status after surgery [17].

In the immediate postoperative phase, nutritional therapy may only minimally counteract muscle catabolism, even providing the energy for optimal healing and recovery. Surgical trauma and possible infection can impair the restoration of lean mass. Severely malnourished patients may exhibit an adynamic form of sepsis. in these situations nutritional therapy will not maintain lean mass but may support the adequate stress response, thus promoting recovery.

In mildly malnourished patients, short-term (7–10 days) preoperative nutritional treatment has to be considered. In severely malnourished patients, nutritional conditioning and mild aerobic exercise must be combined for almost 15 days: nutritional support and physical exercise are prerequisites to rebuild peripheral protein mass [17]. After surgery with infectious complications, at least 6 weeks and sometimes longer may be required to restore a metabolic and nutritional state [39].

The 2017 ESPEN guidelines strongly recommend nutritional intervention before major surgery in case of severe nutritional risks or overt malnutrition, administrated for a minimum of 10 to 14 days, even at the price of delaying surgical intervention [2].

Some studies have shown that an oral diet should be the first choice of feeding in these patients. although the evidence is still of low quality being based on meta-analyses and guidelines [40].

The 2017 ESPEN guidelines underline the importance of physiologic knowledge to personalize nutrition in clinical practice: clinical observation with a “metabolic” view remains mandatory [2].

Nutritional counselling and oral nutritional support (ONS) as a first intervention are not debatable; in case oral feeding is not feasible, the choice of nutrient supply routes are influenced by several factors such as the clinical situation of a patient, resources, and a patient’s will. All of the above factors explain why different practices are carried out within different centers [41].

#### 3.3.1. Early Oral and Enteral Nutrition

Enteral nutrition (EN) is currently the primary recommended approach of nutritional therapy [2] as it can shorten hospital stay, has fewer complications, and reduces costs compared with parenteral nutrition (PN) [42]. According to the ESPEN guidelines [2], the enteral route is contraindicated in case of bowel obstruction, ischemia or hemorrhage, shock and high output fistula.

Early postoperative enteral nutrition is associated with significant reductions in total complications compared with traditional postoperative feeding practices and can positively influence the incidence of mortality and postoperative complications, promoting an earlier resumption of bowel function, and a shortage of hospitalization [43].

In addition to providing nutrients, EN helps to maintain the trophism of the intestinal mucosa, bile secretion, and gastrin production. It can stimulate the intestinal blood’s flow, and can in some way prevent the proinflammatory effect of lymphocytes Th1 and the production of proinflammatory cytokines (IL-1, TNF-b) [44].

Early oral nutrition after surgery can be initiated immediately if tolerated. Early normal food or EN, including clear liquids on the first or second postoperative day, does not cause impairment of healing of anastomoses and can lead to significantly shortened hospital length of stay [45].

The effect of EN on the outcome after surgery in well-nourished patients has not been assessed consistently. The ESPEN guidelines working group reviewed 35 controlled trials [46]. Early EN was compared with normal food, administration of crystalloids, and PN. Compared with PN, early EN decreased postoperative infection rate in undernourished GI cancer patients, but not in those who were well nourished [45,46].

#### 3.3.2. Immune-Modulating Nutrition

In recent years, arginine, glutamine, branched chain amino acids, n-3 fatty acids, and nucleotides which may increase immunity by modulating inflammatory responses or enhancing protein synthesis after surgery have been proposed in addition to standard nutritional formulas. The potential to influence the activity of the immune system by specific nutrients is defined as immunonutrition.

The use of immunonutrition is recommended in the perioperative period for 5 to 7 days in major upper abdominal surgeries, head and neck cancer, and severe trauma [47]. A meta-analysis from Song et al. [48] and Osland et al. [49] confirmed the benefits for the peri- and postoperative use; the usefulness of immune-enriched supplements has not been proven in the preoperative period, as no significant differences were found either in the complication rate or in functional capability and body weight. As the cost of such formulas is generally high and their efficacy might be reduced by perioperative factors, patients who would benefit from immunonutrition must be carefully selected [50].

A 2017 study [51] evaluated the potential benefits of different combinations of immunonutrients in major abdominal surgery. A total of 7116 patients were evaluated. Taking all trials into account, immunonutrients compared with control groups seemed to reduce hospital stay, overall and infectious complications with a low-moderate grade of evidence. However, non-industry-funded trials reported no positive effects compared with those that are industry funded, this bias lowering confidence in the reported results. A multicenter randomized clinical trial comparing Perioperative standard oral nutrition supplements versus immunonutrition in normo-nourished patients undergoing colorectal resection in an enhanced recovery (ERAS) protocol has shown that patients treated with immunonutrients preoperatively and postoperatively had fewer infectious complications than the control group. However, these results need a more solid confirmation as the number of cases was insufficient to make robust conclusions and the methodology was possibly biased [52].

#### 3.3.3. Parenteral Nutrition

PN for well-nourished or mildly depleted patients has been shown to provide a slight clinical benefit and significant morbidity improvements. In severely malnourished patients it can, on the contrary, improve the nitrogen balance and to reduce the incidence and severity of infectious complications. Other studies described the reduction of non-infectious severe complications in severely malnourished patients from 42% to 5% [42].

PN is beneficial in undernourished surgical patients in whom EN is not feasible or not tolerated, because of postoperative complications impairing gastrointestinal function or in case of inability to receive or absorb adequate amounts of oral/enteral feeding for at least 7 days.

In a case of a malnourished patient screened before surgery. a 7–10-day course of preoperative nutrition is recommended [46]. PN has not been shown to decrease morbidity or mortality even if it may lower complication rates in the postoperative period [53].

PN is recommended only when patients are unable to consume food for 7–10 days if previously well-nourished, or for 5–7 days in those previously screened as malnourished prior to surgery [46]. Routine administration of PN postoperatively, has not been shown to have beneficial effects and may be responsible of a 10% increase in the complications rate [54].

Excessive carbohydrate infusion can cause hyperglycemia which can result in adverse outcomes if uncorrected. Additionally, volume overload in individuals with a marginal cardiopulmonary reserve can cause respiratory complications. Overfeeding with hypertonic dehydration and metabolic acidosis is another concern with PN, especially in patients at extreme ages [55].

The effect of PN on the prognosis of surgical patients in comparison with oral/enteral standard nutrition is still controversial. The ESPEN guidelines working group [46] revised 20 randomized studies in which PN was compared with EN, or with a normal hospital diet. No significant difference was found in 8 of the 15 studies, which led most authors to favor EN because of its lower costs.

Heyland et al. incorporated in a meta-analysis 27 studies concerning the influence of PN on morbidity and mortality in surgical patients [56] observing a lower complication rate in malnourished patients receiving PN without any influence on the mortality rate A further meta-analysis by Braunschweig et al. comparing EN with PN, (both surgical and non-surgical) [57] showed a significantly lower risk of infection in the oral/enteral nutrition group. In malnourished patients, however, PN resulted in significantly lower mortality with a trend towards lower rates of infection. Peter et al. [58] found lower infection rates and a shortened length of hospital stay in the enterally fed patients group compared to that of PN. Harvey et al. performed a multicenter randomized study to investigate EN and PN outcomes in 2388 critically ill patients. No difference in mortality, infectious complication rate, and hospital length of stay was observed between the two groups [59].

### 3.4. Prevention and Treatment of Catabolism and Malnutrition

To treat catabolism and malnutrition which are often related to malignancies, and to bring the patients to surgery as fit as possible, preoperative conditioning has been proposed [60,61].

High-risk surgical cancer patients have a low lean mass due to undernutrition and/or cancer-related muscle catabolism: the impairment of aerobic capacity negatively affects functional reserve [62]. Recent studies suggest that several cytokines produced by the tumor or secreted by the host in response to the tumor induce muscle hypercatabolism [63]. In cancer patients, sarcopenia has been of pro-inflammatory cytokines and leptin, which influence insulin resistance and energetic metabolism. The combination of visceral obesity and sarcopenia increases the likelihood of postoperative mortality and affects the prognosis of postoperative complications [64,65].

Optimizing body composition and enhancing oxygen uptake ability before surgery may contribute to improved postoperative outcomes; this can be obtained only with a multimodal approach [12] including nutritional, psychological and physical exercise intervention, beneath the new concept of “prehabilitation”. In a randomized trial concerning colorectal surgery [66], a prehabilitation program enhanced the patient’s functional recovery reducing postoperative morbidity, and shortening hospital stay following colorectal surgery. Similar results were achieved by two recent meta-analyses dealing with patients undergoing cardiac and abdominal surgery [67,68]. A randomized, blind controlled trial carried out in high-risk patients undergoing major GI surgery [69] showed that a personalized prehabilitation program can enhance aerobic capacity and reduce postoperative complications. Sarcopenia assessment can help identify cancer surgery patients at risk of poor outcomes. CT and magnetic resonance imaging (MRI) are considered the gold standard methods of assessing body composition in research, and image analysis software can be used to obtain reliable body composition measures for sarcopenia assessment. Alternative methods of assessment are bioelectrical impedance analysis (BIA) and dual-energy X-ray absorptiometry (DXA). BIA results can be influenced by fluid status, and this method is not ideal for adiposity assessment. DXA is not widely available and can misclassify body composition in individuals with high levels of water and fibrous tissue. Current evidence is insufficient to support these alternative means for sarcopenia diagnosis [70].

The most effective interventions against sarcopenia are physical exercise and enhanced oral protein intake. Pharmacological therapies for sarcopenia (testosterone, androgen receptor modulators, ghrelin agonists, myostatin inhibitors, ACE inhibitors) have been evaluated, but they are generally less effective than postulated (idem).

Sarcopenic, elderly, frail, and cancer patients could benefit more from prehabilitation than other patient populations: however, since prehabilitation programs are different for the duration of the intervention (3–6 weeks) and type of exercise training, future studies should contribute to standardize the preoperative pathway.

## 4. Discussion

Malnourished patients undergoing surgery have a dramatically increased risk of postoperative complications, leading to elongation of hospital stay and costs, decreased quality of life, and risk of nosocomial complications.

Assessing the nutritional status of every patient, in particular in the care of elderly and cancer patients, is a crucial feature of the therapeutic pathway in order to optimize every strategy.

Among the various tools in use to assess the nutritional status, the PG-SGA seems to be the most reliable and complete.

For patients recognized as malnourished, nutrition preoperative therapies are indicated, paying attention to selecting the correct approach for the right patient. Among the most popular strategies, enteral nutrition seems to have the best risk–benefit ratio.

Early oral nutrition after surgery is preferable, when it feasible, and should be administered in all the patients undergoing elective major surgery.

## 5. Conclusions

The impact of artificial nutrition on postoperative complications is nowadays undisputed in malnourished patients or at risk of undernutrition. Preoperative malnutrition is related to a prolonged hospital stay, greater postoperative complications, a greater risk of re-hospitalization, and a greater incidence of postoperative death. To assess the nutritional status and identify the risk of malnutrition and consequent surgical morbility, screening tools based on BMI and body weight loss must be employed. All surgical oncologic patients should follow a multidisciplinary evaluation process for optimal management. Promising results are coming from prehabilitation programs. Early postoperative enteral nutrition has been shown to be safe and viable for surgical oncology patients. Compared to PN, EN improves the postoperative nutritional status of patients more efficiently, playing an important role in restoring the intestinal barrier function in the postoperative phase, reducing the incidence of postoperative infections, and lowering the cost of hospitalization. The real efficacy of immunonutrition is still not clearly demonstrated.

## Figures and Tables

**Figure 1 healthcare-08-00559-f001:**
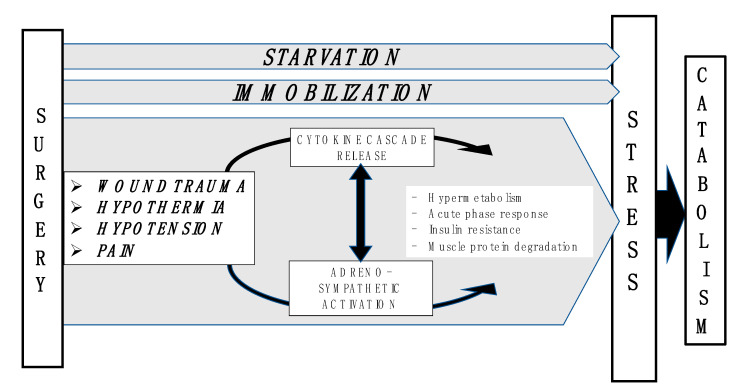
The main causes of metabolic changes consequent to surgery.

**Figure 2 healthcare-08-00559-f002:**
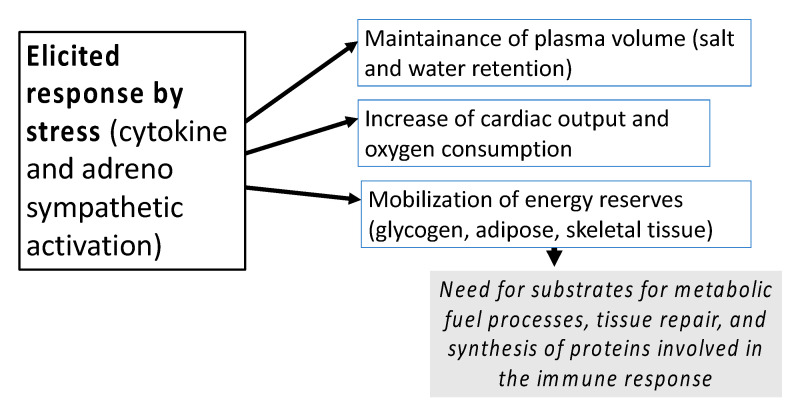
The consequences of the elicited stress reaction.

**Table 1 healthcare-08-00559-t001:** Indications to nutritional interventions.

Factors	Influence on Outcome	Treatment
Preoperative malnutrition	Increased infectious complications Delayed recovery	Preoperative nutrition
Long starving	Increased infectious complications Delayed recovery	Pre- and intraoperative nutrition: prefer enteral route to parenteral nutrition due to associated morbidity
Intraoperative surgical stress	Catabolism, immunosuppression, organ dysfunction	Early enteral nutrition, minimally invasive surgery
Catabolism/muscle loss	Increase of overall morbidity Increase of fatigue Delayed recovery	Active rehabilitation Early oral nutrition
Severe complications (ileus, high output fistula, intestinal bleeding hemorrhage)	Contraindications to enteral nutrition	Parenteral nutrition

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
