# Peer review of "Impact of Artificial Nutrition on Postoperative Complications"

_healthcare, 2020, doi:10.3390/healthcare8040559_

Round 1

Reviewer 1 Report

The article "Impact of Artificial Nutrition on Postoperative Complications" presents a review that requires some improvements to be published in Healthcare:

-In my opinion this paragraph is essential in the context of the article: "To assess and optimize the nutritional status of patients undergoing surgery, it is essential to understand how the metabolism changes during injury, and why a poor nutritional status is a risk factor for postoperative complications" Therefore, a schematic figure should be included that helps the reader to pay attention to this central aspect.

-Material and Methods is mentioned:
"A PubMed / Medline search was performed using search terms" nutrition and surgery, "" nutrition and surgical oncology "," effects of nutrition on surgical complications, "" nutritional screening in surgical oncology patients ". Then line 77 indicates" Patients undergoing gastrointestinal surgery a ... "please clarify in materials and methods precisely the scope.

-It should be briefly defined for non-expert readers in which conditions in general nutritional interventions are indicated, describe them briefly as well and please include a figure that connects these types of interventions with postoperative complications.

Author Response

reviewer 1

The article "Impact of Artificial Nutrition on Postoperative Complications" presents a review that requires some improvements to be published in Healthcare:

Q-In my opinion this paragraph is essential in the context of the article: "To assess and optimize the nutritional status of patients undergoing surgery, it is essential to understand how the metabolism changes during injury, and why a poor nutritional status is a risk factor for postoperative complications" Therefore, a schematic figure should be included that helps the reader to pay attention to this central aspect.

  • two figures has been added

Q-Material and Methods is mentioned:
"A PubMed / Medline search was performed using search terms" nutrition and surgery, "" nutrition and surgical oncology "," effects of nutrition on surgical complications, "" nutritional screening in surgical oncology patients ". Then line 77 indicates" Patients undergoing gastrointestinal surgery a ... "please clarify in materials and methods precisely the scope.

A- this observation has been followed and the line 77 corrected

Q-It should be briefly defined for non-expert readers in which conditions in general nutritional interventions are indicated, describe them briefly as well and please include a figure that connects these types of interventions with postoperative complication

  • A table has been added for more clarity

Reviewer 2 Report

Comments to Authors:

Title: Impact of Artificial Nutrition on postoperative Complications

This paper describes the effectiveness of perioperative immunonutrition in patients with high risk for postoperative infections.

This paper including relatively large number of references so the results may be highly reliable.

But there are some questions and the author is requested to add the descriptions according to comments as below.

1) Immunonutrition within enhanced recovery after surgery

There is no detailed description about immunonutrition within enhanced recovery after surgery.The author should add the descriptions of immunonutrition within enhanced recovery after surgery.

2) Sarcopenia

Sarcopenia is also one of risk factors for postoperative infections.The author should add the descriptions about assessment and nutritional interventions in sarcopenic patients.

Author Response

reviewer 2

This paper describes the effectiveness of perioperative immunonutrition in patients with high risk for postoperative infections.

This paper including relatively large number of references so the results may be highly reliable.

But there are some questions and the author is requested to add the descriptions according to comments as below.

Q- Immunonutrition within enhanced recovery after surgery. There is no detailed description about immunonutrition within enhanced recovery after surgery.The author should add the descriptions of immunonutrition within enhanced recovery after surgery.

A- A description of immunonutrition within enhanced recovery after surgery has been added

Q-  Sarcopenia. Sarcopenia is also one of risk factors for postoperative infections.The author should add the descriptions about assessment and nutritional interventions in sarcopenic patients

  • A description about assessment and nutritional interventions in sarcopenic patients has been added

Round 2

Reviewer 1 Report

Accept

Reviewer 2 Report

Comments to Authors:

Title: Impact of Artificial Nutrition on postoperative Complications

This paper describes the effectiveness of perioperative immunonutrition in patients with high risk for postoperative infections.

This paper including relatively large number of references so the results may be highly reliable.

The authors added the detailed descriptions in the paper according to comments I requested.

But there are some questions and the author is requested to add the descriptions according to comments as below.

1. The style of references

The style of references was different in some parts.

Ref, 33.   Am J Clin Nutr. 417 2008;(87:1678–1688).

Ref, 38.   Br J Surg. 2010;(97:92-97).

Ref, 41.   Int J Surg. 439 2015;(18:7-13).

The author should change the style of references according to the paper instruction.